# MambaExtend: A Training-Free Approach to Improve Long-Context Extension of Mamba

**Seyedarmin Azizi**[u]†**, Souvik Kundu**[i]†**, Mohammad Erfan Sadeghi**[u]**, and Massoud Pedram**[u]

[u]University of Southern California, Los Angeles, USA
[i]Intel Labs, USA
† Equal contribution authors
{seyedarm, sadeghim, pedram}@usc.edu, souvikk.kundu@intel.com

## ABSTRACT

The inherent quadratic complexity of the attention mechanism in transformer models has driven the research community to explore alternative architectures with sub-quadratic complexity, such as state-space models. Mamba has established itself as a leading model within this emerging paradigm, achieving state-of-the-art results in various language modeling benchmarks. However, despite its impressive performance, Mamba's effectiveness is limited by its pre-training context length, resulting in a pronounced degradation when the model is tasked with handling longer contexts. Our investigation reveals that Mamba's inability to generalize effectively to long contexts is primarily due to the out-of-distribution (OOD) discretization steps. To address this critical limitation, we introduce ***MambaExtend***, a novel framework designed to significantly enhance the context extension capabilities of Mamba. Specifically, MambaExtend leverages a ***training-free*** approach to calibrate *only* the scaling factors of discretization modules for different layers. We demonstrate both gradient-based and gradient-free zeroth-order optimization to learn the optimal scaling factors for each Mamba layer, requiring orders of magnitude fewer updates as opposed to the parameter fine-tuning-based alternatives. Using this approach, we achieve a training-free context extension of up to $32\times$, expanding the context from 2k to 64k tokens with minimal increases in perplexity. In contrast to existing fine-tuning methods, MambaExtend selectively calibrates the scaling factors, requiring up to $\sim 5.42 * 10^6 \times$ fewer parameter updates and incurring up to $3.87\times$ lower peak memory usage, while delivering comparable or superior long-context performance across multiple tasks. Codes and checkpoints are available here[1].

## 1 INTRODUCTION

Despite the widespread applications of transformer (Vaswani, 2017) based large language models (LLMs) (Touvron et al., 2023), their quadratic compute and memory demand with sequence length has enforced research for emerging alternative architectures. For example, works including Linformer (Wang et al., 2020) and Longformer (Beltagy et al., 2020) presented different approaches to approximate attention to reduce the quadratic memory cost. Other works (Kitaev et al., 2020) leveraged locality-based hashing to avoid attention computation. Recently, state-space models (SSMs) (Gu et al., 2022; 2020) have emerged as an alternative to attention-based models, offering a

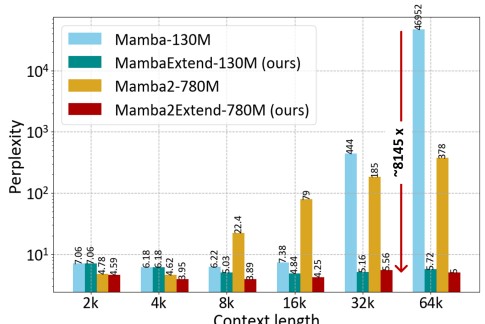

Figure 1: Long-context understanding on Pile. Compared to the pre-trained alternatives, MambaExtend provides up to $\sim 8145\times$ improvement in perplexity score, via a **training-free** calibration.

---

[1]https://github.com/ArminAzizi98/LongContextMamba

different approach to handling long sequences at sub-quadratic complexity. Unlike transformers, SSMs are grounded in continuous-time dynamics and offer the potential to handle much longer sequences without blowing out the memory and compute demand. Mamba (Gu & Dao, 2023; Dao & Gu, 2024), a popular SSM variant built leveraging the selective state-space layers (S6), has shown impressive performance on various NLP, image, and medical genomics benchmarks (Schiff et al., 2024). The key advantage of Mamba stems from the sub-quadratic compute complexity of theoretically grounded linear RNN layers.

LLMs for long-context understanding have recently found many useful applications, including summarizing long documents and answering long questions (Chen et al., 2023c). However, transformer-based LLMs that are pre-trained on fixed-length contexts yield lower generative performance when used on longer sequences during inference time (Chen et al., 2024; 2023b). This shortcoming of the transformers is tied to the inability of the positional embedding to generalize well on longer sequences, causing such sequences to appear as out-of-distribution (OOD) sequences (Chen et al., 2023c; Jin et al., 2024). Interestingly, Mamba models, despite their theoretical ability to capture global interactions, also fail to generalize to long sequence or context lengths (Ben-Kish et al., 2024). This phenomenon has been tied to the Mamba model's implicit bias to a limited effective receptive field (ERF) governed by the training data sequence length (Ben-Kish et al., 2024).

For transformer-based LLMs, the OOD sequence length generalization has been explored extensively, including fine-tuning to longer sequences (Chen et al., 2023c) and allowing sophisticated modification to the transformer's positional embedding (Jin et al., 2024; Ding et al., 2024; Golovneva et al., 2024). Unfortunately, such solutions are not directly applicable to Mamba models. This is primarily due to the absence of an explicit positional embedding for Mamba models to generalize. Moreover, unlike transformers, the potential root cause of Mamba's performance deterioration for long sequence processing is yet to be discovered.

A contemporary work, namely DeciMamba (Ben-Kish et al., 2024), has presented a selective token *decimation* strategy to reduce the number of tokens to be processed per layer. This approach potentially increases the model's ERF, enabling better long-context information flow and understanding. However, DeciMamba requires a memory- and compute-intensive fine-tuning of the model, resulting in significant time and effort to perform parameter updates of the pre-trained model. Thus, such an approach does not scale to larger models, especially for limited memory or computing resources.

**Our Contributions.** To mitigate the aforesaid issues, we first investigate the impact of OOD long-context extension on the discretization steps of Mamba ($\Delta_t$ values).[2] We have empirically observed two key findings: (1) minimizing the sum of $\Delta_t$'s across all tokens can enhance generalization when increasing context length during inference, and (2) applying a fixed down-scaling to $\Delta_t$'s generally does not yield optimal generalization. Building on this insight, we introduce ***MambaExtend***, a framework designed to extend Mamba's context length **without any re-training** of the model weights. MambaExtend utilizes a calibration function (CF) to optimize sizes of the discretization steps ($\Delta_t$) across various Mamba layers by incorporating a learnable scaling factor for each layer's $\Delta_t$. This CF enables the proposed scaling parameters for $\Delta_t$ to be learned while keeping the model weights fixed at their pre-trained values, significantly reducing both memory requirements and the number of updatable parameters. Moreover, we introduce a zeroth-order (ZO) optimization-based CF to calibrate using only forward passes, potentially yielding further savings in memory and computation. Specifically, we used ZO methods based on the simultaneous perturbation stochastic approximation (SPSA) (Spall, 1992) to update the scaling factors. As shown in Fig. PPL by up to $\sim \mathbf{8145\times}$, as evaluated on context lengths of up to $64$k.

To show the efficacy of MambaExtend, we performed extensive experiments on perplexity evaluation, LongBench, and long-context retrieval tasks with both Mamba and Mamba2 variants. For example, on PG19, only via ZO-based scaling factor update, MambaExtend can improve the context length extension ability of a pre-trained model from 2k to $64$k, while not incurring any significant perplexity (PPL) increase. Compared to DeciMamba, we yield up to **40.6%** reduced PPL while requiring up to $\sim \mathbf{5.42 * 10^6 \times}$ fewer parameter update with up to $\mathbf{3.87\times}$ lower peak-memory demand.

---

[2]$\Delta_t$ represents the step size that transforms continuous-time parameters into the corresponding discrete state-space variables.

## 2 PRELIMINARIES

### 2.1 THE S6 LAYER AND MAMBA

At its core, each Mamba block utilizes the selective SSM (S6) layer (Gu & Dao, 2023), which is specifically designed to handle sequential data by preserving structured state dynamics across the input sequence.

**The S6 layer**: Using a **linear recurrent system** with the hidden state $h_t$, input $z_t$, and output $o_t$ at discrete time instant $t$, the S6 layer's sequence generation and state update can be simplified as:

$$h_t = \bar{A}h_{t-1} + \bar{B}z_t, \, o_t = Ch_t \tag{1}$$

The P-length sequence of a representative channel is given as $Z = \{z_1, z_2, \cdots, z_P\}$, $\bar{A} \in \mathbb{R}^{N \times N}$, $\bar{B} \in \mathbb{R}^{N \times 1}$, and $C \in \mathbb{R}^{1 \times N}$ are discrete time-variant system, input, and output matrices, respectively, governing the discrete state transitions and output sequence generation. The S6 layer produces the 'per-time' ($t$) discrete time-variant matrices from input and "continuous parameters" as:

$$\bar{A}_t = \exp(\Delta_t A), \, \bar{B}_t = \Delta_t B_t \text{ where } \Delta_t = \text{SFT}(\Delta_{t_{proj}}(z_t)), \, B_t = W_B(z_t), \, C_t = (W_C(z_t))^T \tag{2}$$

Here, $z_t \in \mathbb{R}^D$ with $D$ being channel dimension and $\Delta_t$ be the discretization step used at time $t$. $\Delta_{t_{proj}}$, $W_B$, and $W_C$ are linear projection layers. SFT and exp represent the *softplus* and pointwise *exponential* operation, respectively. After the discretization step, the S6 layer's input-output behavior via time-unrolling can be described as:

$$O = \alpha Z \text{ with } \alpha_{i,j} = C_i \left( \prod_{k=j+1}^{i} \bar{A}_k \right) \bar{B}_j \tag{3}$$

Thus, for a context length of $P$, the entire output $O = \{o_1, o_2, .., o_P\}$ is computed as follows:

$$\begin{pmatrix} o_1 \\ o_2 \\ \vdots \\ o_P \end{pmatrix} = \begin{pmatrix} C_1\bar{B}_1 & 0 & \cdots & 0 \\ C_2\bar{A}_2\bar{B}_1 & C_2\bar{B}_2 & \cdots & 0 \\ \vdots & \vdots & \ddots & \vdots \\ C_P \prod_{k=2}^{P} \bar{A}_k \bar{B}_1 & C_P \prod_{k=3}^{P} \bar{A}_k \bar{B}_2 & \cdots & C_P\bar{B}_P \end{pmatrix} \begin{pmatrix} z_1 \\ z_2 \\ \vdots \\ z_P \end{pmatrix} \tag{4}$$

This matrix formulation shows that each output $o_i$ is a weighted sum of the inputs $z_1, z_2, \ldots, z_P$, with the weights determined by the state-space matrices $\bar{A}$, $\bar{B}$, and $C$. The model can thus integrate information across different time steps while maintaining computational efficiency. This matrix resembles the attention score map in transformer-based models (Ali et al., 2024). In other words, S6 layers may be interpreted as data-controlled linear operators.

Notably, as these matrices are dynamically adjusted based on the input sequence, they enable the model to efficiently capture temporal dependencies across various time steps. This approach allows Mamba to maintain computational complexity that scales linearly with the context length.

**Mamba block.** One of the critical aspects of Mamba's architecture is how a Mamba block relates its input sequence $X = (x_1, x_2, \ldots, x_P)$ to its output sequence $Y = (y_1, y_2, \ldots, y_P)$ with $P$ corresponding to the sequence or context length. The relationship between the input and output of the Mamba block is expressed through a time-varying SSM described below:

$$G = \sigma(W_{gate\_proj}X), \, Z = \text{Conv1D}(W_{in\_proj}X) \tag{5}$$

$$O = \text{S6}(Z), \, Y = O \odot G \tag{6}$$

Here, $G$ is a gating function derived from a linear transformation of the input sequence $X$ followed by a SILU function, $\sigma$. The element-wise multiplication $\odot$ between $G$ and $O$ allows the model to selectively emphasize or attenuate parts of the input to focus on relevant input information. The input $Z$ to the S6 is a linearly transformed version of the original input $X$ followed by a 1D convolution.

As demonstrated in these equations, the relationship between the last token $o_P$ and the first token is governed by the term $\alpha_{P,1} = C_P \prod_{k=2}^{P} \bar{A}_k \bar{B}_1 = C_P \exp(A \sum_{k=2}^{P} \Delta_k)\bar{B}_1$. This means that the exponent of summed $\Delta_t$ determines the impact of the first token in the generation of the $P^{th}$ token.

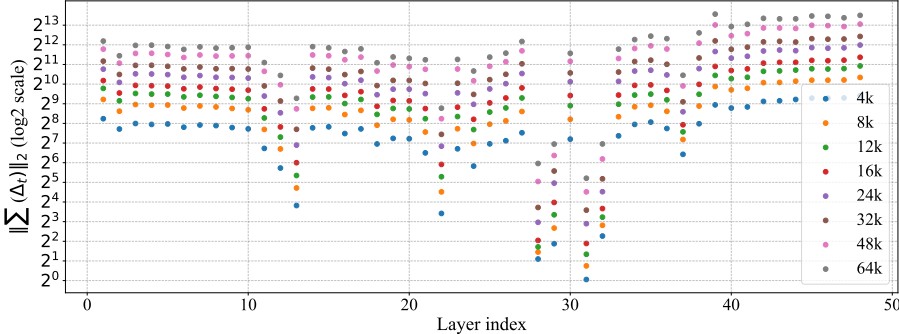

Figure 2: Layer-wise behavior of $\sum(\Delta_t)$ for different context length during test-time. We used the Pile dataset on Mamba 1.4B for the evaluation.

## 3 MOTIVATIONAL CASE STUDIES

**The behavioral change of** $\Delta_t$. We first investigate the behavior of the accumulated discretization matrix $\Delta_t$ in the pre-trained Mamba-1.4B model when exposed to inputs of different context lengths. Using 100 samples from Pile, for each Mamba layer, we compute the $\|(\sum_{t=1}^{P'} \Delta_t)\|_2$ for different evaluation context lengths $P'$, where $\|.\|_2$ represents the $l_2$-norm of a tensor. We plot this analysis in Fig. 2, which reveals how the accumulation of $\Delta_t$ scales with increasing context length. Specifically, Fig. 2 discloses that for each layer of the model, the magnitude of $\sum \Delta_t$ increases with the increase in context lengths $P'$. According to Equations 3 and 4, and given that all entries of $A$ are always negative (Gu & Dao, 2023), we observe that the negative sum of $\Delta_t$ appears as the exponent in the exp function. Consequently, the term $\exp(-\sum \Delta_t)$ effectively governs the decay of influence from any previous token. A larger value of $\Delta_t$ results in greater forgetfulness, decreasing the model's reliance on earlier tokens. In contrast, smaller $\Delta_t$ values enable the model to retain information from more distant tokens. Therefore, $\exp(-\sum_{t=n}^{P'} \Delta_t)$ can be interpreted as a parameter that potentially regulates the retention level for the $n$-th input to compute the token at $P'$.

**Influence of scaled** $\Delta_t$. For transformer-based LLMs, a popular method for addressing the out-of-distribution (OOD) context length $P' > P$ (where $P$ represents the training context length) is positional interpolation (PI) (Chen et al., 2023a). The PI method accomplishes this by multiplying the token index value in RoPE by $\frac{P}{P'}$. This rescaling ensures that the positional indices remain within a valid range, effectively mitigating the OOD problem associated with longer contexts without retraining.

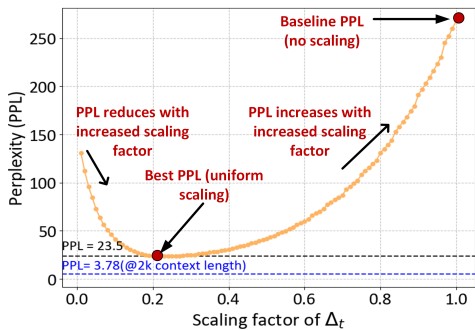

Figure 3: Impact of different values of uniform $\Delta_t$ scaling on the perplexity (PPL) evaluation metric.

Inspired by this, we propose a straightforward approach for Mamba to address the accumulated out-of-distribution (OOD) discretization steps by scaling the discretization matrix $\Delta_t$ by a fixed scalar value $s \leq 1$ across all model layers. This method aims to mitigate the OOD effects associated with longer context sizes. We utilized a pre-trained Mamba 1.4B to validate this approach, conducting a grid search over various values of $s$. We then evaluated the model's performance on the test set of the Pile dataset (Gao et al., 2020) for an evaluation context length of 32k tokens, reporting the average perplexity. The results, presented in Fig. 3, demonstrate that **scaling** $\Delta_t$ **can significantly reduce the model's PPL from approximately 268 to around 23.5**. However, the findings also indicate that the relationship between the choice of scaling value and performance improvement is not straightforward. As shown in Fig. 3, while increased scaling helps reduce the perplexity at lower values of (s), the PPL rises after reaching a certain threshold. This complex interplay encourages us to investigate the model's capacity to learn the optimal scaling. Additionally, **this uniform scaling factor cannot restore the model's performance for longer contexts to the level observed at its**

**pre-trained context length**. For instance, the model achieves a PPL of 3.7 for a 2k context length, which remains significantly lower than the best PPL obtainable through uniform scaling.

**Variable impact of $\Delta_t$ on different layers.** Another important observation in Fig. 2 is that for a given test-time context length $P'$, different layers of the model produce significantly different $\sum \Delta_t$ values (even when viewed on a logarithmic scale). This underscores the point that **each layer should not employ the same scaling factor to reduce the impact of $\Delta_t$**. This observation motivates us to implement a heterogeneous (layer-specific) scaling mechanism across the various layers of the model to effectively address the OOD $\sum \Delta_t$.

## 4 MambaExtend Methodology

Motivated by the need to mitigate the OOD effects, we introduce MambaExtend, a *training-free* method for scaling the discretization steps of each layer. For an $L$-layer Mamba model, our primary objective is to determine the optimal scaling factors for each layer, denoted as $s_1, s_2, \ldots, s_L$, which will be used to adjust the discretization matrix $\Delta_t$. Note that for a layer $i$, $s_i \in \mathbb{R}^m$, where $m = 1$ indicates that $s_i$ is a scalar, and $m > 1$ indicates that $s_i$ is a vector. Without loss of generality, for $m = 1$, the discretization adjustment can be expressed as $\Delta'_{t_i} = s_i \Delta_{t_i}$, with $\Delta'_{t_i}$ applied during inference. The goal is to calibrate the newly introduced learnable parameters $s_i$ for all $i \in 1, \ldots, L$ in a way that is both memory- and compute-efficient, and *does not* involve any additional training or fine-tuning of the model parameters. These constraints will enable such calibration to be feasible on resource-limited edge devices.

---

**Algorithm 1** MambaExtend Algorithm

---

1: **Input:** An $L$-layer Mamba model parameterized by $\mathcal{M}$, set of calibration samples $\mathcal{C}$, calibration function `CF`
2: **Output:** Scaling factors $\mathbf{S} = [s_1, s_2, ..., s_L]$, where $s_i \in \mathbb{R}^m$
3: **for** $i \leq L$ **do**
4: $\quad s_i \leftarrow$ `init`$(U(0,1))$
5: **end for**
6: `freeze`$(\mathcal{M})$
7: $\mathbf{S} \leftarrow$ `CF`$(\mathbf{S}, \mathcal{C}, \mathcal{M})$
8: **return S**

---

Algorithm 1 outlines the MambaExtend framework, which takes a pre-trained Mamba model as input, along with a small set of calibration samples from the target task and a specialized function known as the **calibration function** (`CF`). As its name implies, `CF` calibrates the learnable scaling factors. Importantly, unlike DeciMamba, which allows fine-tuning of the weights, MambaExtend keeps the model weights fixed to their pre-trained values (as indicated in Line 6 of Algorithm 1) throughout the calibration process. This approach makes MambaExtend significantly more compute- and memory-efficient compared to DeciMamba.

**Calibration via back-propagation (`CF`$_{BP}$).** Gradient-based backpropagation is a widely used optimization method for updating the free (unfrozen) parameters on a calibration set. However, to minimize computational and memory overhead, we ensure parameter efficiency by restricting updates to the scaling factors $\mathbf{S}$ only. Algorithm 2 summarizes the `CF`$_{BP}$ algorithm for finding the optimal scaling factors. We utilize Adam as the optimizer for backpropagation (as noted in Line 4 of Algorithm 2). The `Evaluate()` function in Line 6 computes the loss of the model, which is parameterized by frozen weights and the learnable scaling factors $\mathbf{S}$.

**Calibration via zeroth-order optimization (`CF`$_{ZO}$).** Zeroth-order optimization (Spall, 1992; Malladi et al., 2023b) offers an efficient yet noisier method for calibration, as it relies solely on forward passes to approximate gradients. Algorithm 3 outlines the process for optimizing the scaling factors $\mathbf{S}$ in `CF`$_{ZO}$. Specifically, this is a multi-iteration process in which, at each iteration, the scaling factors are randomly perturbed using a random variable $\delta$ sampled from a Rademacher distribution. The magnitude of the perturbation and the learning rate for the updates are controlled by the hyperparameters $c$ and $\eta$, respectively. We employ the two-sided variant of the simultaneous perturbation stochastic approximation method (SPSA) (Spall, 1992), which obtains gradient approximations by

---

**Algorithm 2** $\texttt{CF}_{BP}$ Algorithm

---

1: **Input:** An $L$-layer Mamba model parameterized by frozen weights $\mathcal{M}$, set of calibration samples $\mathcal{C}$, the initialized scaling factors $\mathbf{S}$
2: **Input:** Learning rate $\eta$, number of iterations $K$
3: **Output:** Learned Scaling factors $\mathbf{S} = [s_1, s_2, ..., s_L]$, where $s_i \in \mathbb{R}^m_+$
4: $\texttt{optimizer} = \text{Adam}(\mathbf{S}, \eta)$
5: **for** $k \leq K$ **do**
6: $\quad \mathcal{L} = \texttt{Evaluate}(\mathcal{M}_{\Delta_t \times \mathbf{S}}, \mathcal{C})$
7: $\quad \mathcal{L}.\texttt{backward()}$
8: $\quad \texttt{optimizer.step()}$
9: $\quad \mathbf{S} \leftarrow \mathbf{S}.\texttt{clamp}(\min = 0.001)$ # make sure scaling factors remain positive
10: **end for**
11: return $\mathbf{S}$

---

**Algorithm 3** $\texttt{CF}_{ZO}$ Algorithm

---

1: **Input:** An $L$-layer Mamba model parameterized by $\mathcal{M}$, set of calibration samples $\mathcal{C}$, the initialized scaling factors $\mathbf{S}$
2: **Output:** Learned scaling factors $\mathbf{S} = [s_1, s_2, ..., s_L]$, where $s_i \in \mathbb{R}^m_+$
3: Specify learning rate $\eta$, perturbation magnitude $c$, number of iterations K
4: **for** $k \leq K$ **do**
5: $\quad \delta \in \mathbb{R}^{L \times m} \sim \texttt{Rademacher()}$
6: $\quad \mathbf{S}^+ = \mathbf{S} + c \times \delta, \quad \mathbf{S}^- = \mathbf{S} - c \times \delta$
7: $\quad \mathcal{L}^+ = \texttt{Evaluate}(\mathcal{M}_{\Delta_t \times \mathbf{S}^+}, \mathcal{C}), \quad \mathcal{L}^- = \texttt{Evaluate}(\mathcal{M}_{\Delta_t \times \mathbf{S}^-}, \mathcal{C})$
8: $\quad \hat{\nabla}_{\mathbf{S}} = (\mathcal{L}^+ - \mathcal{L}^-)/(2c\delta)$
9: $\quad \mathbf{S} \leftarrow \mathbf{S} - \eta \hat{\nabla}_{\mathbf{S}}$
10: $\quad \mathbf{S} \leftarrow \mathbf{S}.\texttt{clamp}(\min = 0.001)$ # make sure scaling factors remain positive
11: **end for**
12: return $\mathbf{S}$

---

applying both positive and negative perturbations to the parameters simultaneously. The two-sided SPSA approach yields gradient estimates with lower variance than the one-sided version, thus enhancing accuracy, especially in noisy environments (Spall, 2005).

The convergence of the zeroth-order calibration method, $\texttt{CF}_{ZO}$, is affected by the number of parameters being optimized, specifically the size of $\mathbf{S}$. Classical lower bounds indicate that convergence slows linearly as the number of parameters increases (Nemirovskij & Yudin, 1983; Duchi et al., 2015). Consequently, a natural strategy in our context is to employ the backpropagation-based method, $\texttt{CF}BP$, when optimizing a larger set of parameters in ($\mathbf{S}$), while reserving $\texttt{CF}_{ZO}$ for smaller parameter sets.

Our experiments show that long-context evaluation tasks, based on the perplexity measure, and the LongBench tasks require relatively fewer scaling factors. Specifically, for each layer $s_i \in \mathbb{R}+^m$, a setting of $m = 1$ is sufficient to improve PPL on long-context inputs. Here, $\mathbb{R}+$ represents the set of positive real numbers, as scaling factors cannot take negative values in our case. Any $s_i$ that updates to a negative value is clamped to a very small positive number to ensure this condition in our algorithm. We set $m = D$ for the passkey retrieval task, thereby increasing the number of parameters to be calibrated or updated. We empirically find that for the long-context tasks, $\texttt{CF}_{ZO}$ performs nearly as well as $\texttt{CF}_{BP}$. However, for the passkey retrieval task, we prefer $\texttt{CF}_{BP}$ due to its faster convergence trend compared to the zeroth-order method. In future work, we plan to address the tuning of the zeroth-order approach to achieve a better convergence rate for relatively high parameter counts.

## 5 EXPERIMENTS

This section evaluates the performance and efficiency of our proposed MambaExtend. Specifically, we first describe the models and datasets used for our experiments. We then present extensive

empirical results to outline our findings regarding the long-context performance of the Mamba model variants. We finally discuss the compute, time, and memory requirements for MambaExtend.

## 5.1 EXPERIMENTAL SETUP

**Models and datasets**. To evaluate the performance of MambaExtend, we use both long-context understanding and long-context retrieval ability tasks. For long-context understanding, we use the Pile (Gao et al., 2020) and PG-19 (Rae et al., 2019) datasets and assess the performance of the MambaExtend in terms of perplexity scores at various context lengths. We use Mamba-130M, Mamba-1.4B (Gu & Dao, 2023), and Mamba2-780M (Dao & Gu, 2024) for these evaluations. Additionally, we use the LongBench benchmark (Bai et al., 2023) to evaluate the performance accuracy of the Mamba-1.4B and Mamba2-780M models. In specific, we use seven tasks, namely Qasper (single-document QA), HotpotQA, 2WikiMultihopQA (multi-document QA), TREC, TriviaQA (few-shot learning), LCC, and RepoBench-P (code completion). For the passkey retrieval task, we follow the setup described in (Ben-Kish et al., 2024) and evaluate the performance of the Mamba-130M and Mamba-1.4B models in retrieving a 5-digit code embedded at a random sequence depth within samples from the WikiText-103 dataset (Merity et al., 2016). In our retrieval setup, the input sequence lengths range from 1K to 64K tokens.

**Baseline and SoTA comparison**. We use the pre-trained Mamba (Gu & Dao, 2023) and Mamba2 (Dao & Gu, 2024) models to evaluate the baseline performance as we increase the evaluation context length $P'$. We use DeciMamba (Ben-Kish et al., 2024), a contemporary work that uses memory-intensive fine-tuning to update all the parameters while improving the effective receptive field.

## 5.2 EXPERIMENTAL RESULTS

**Perplexity evaluations on PG-19 and Pile**. To evaluate perplexity (PPL) on the Pile and PG-19, we use twenty calibration samples from the corresponding training set for a given context length. We use these samples to learn the scaling factors in MambaExtend, then evaluate perplexity on the test set for a given context length. As stated earlier for the perplexity evaluation, for each layer $i$, we use a single scaling factor $s_i \in \mathbb{R}_+$ per layer[3], which scales the $\Delta_t$ tensor uniformly for that layer. Therefore, in an $L$-layer Mamba model, we optimize $L$ scaling factors for these datasets. Given the small number of parameters to optimize, we use $\mathrm{CF}_{ZO}$ as the calibration function.

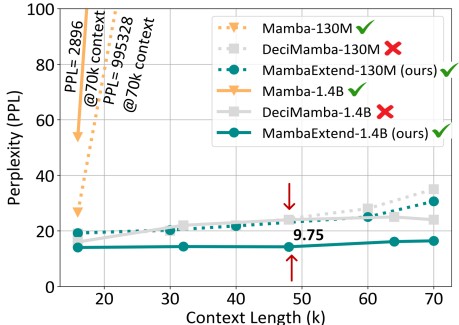

Figure 4: Perplexity comparison on PG-19. The ✓ and ✗ identify the fine-tuning requirements to be false and true, respectively.

Fig. 4 depicts the performance of MambaExtend compared to the pre-trained Mamba variants and DeciMamba. Specifically, at 70k context length, MambaExtend-130M yields a PPL of 30.62, a $\sim\mathbf{32506\times}$ improvement over the baseline counterpart that fails to provide a very high PPL of 995328. Compared to the DeciMamba, it shows consistent improvement with reduced PPL of up to $\sim\mathbf{40.6\%}$.

Table 1 reports the PPL values of MambaExtend models and compares them to those of the pre-trained models on Pile. As shown in the table, MambaExtend, through only minimal calibration, allows the models to maintain their performance even with increasing context lengths. Specifically, MambaExtend can improve the PPL by up to $\sim\mathbf{8145\times}$, showing higher improvement trends at longer contexts.

**LongBench**. LongBench Bai et al. (2023) is a benchmark for bilingual, multitask, and comprehensive assessment of long-context understanding. For MambaExtend, we use seven popular tasks from LongBench. Due to the lack of training data, we used 10 samples from the 4K-8K split of each dataset as calibration data and the remaining samples from the same split to evaluate. We apply the

---

[3]This may be attributed to the relatively more straightforward nature of long-context understanding as opposed to long-context retrieval, since for the latter we need more fine-grain scaling, increasing the number of calibration parameters.

Table 1: Perplexity for Mamba models over different evaluation context lengths on Pile dataset.

| | Mamba-130M | | | | | | Mamba-1.4B | | | | | | Mamba2-780M | | | | | |
|---|---|---|---|---|---|---|---|---|---|---|---|---|---|---|---|---|---|---|
| Context Length | 2k | 4k | 8k | 16k | 32k | 64k | 2k | 4k | 8k | 16k | 32k | 64k | 2k | 4k | 8k | 16k | 32k | 64k |
| Pre-trained Model | 7.06 | 6.18 | 6.22 | 7.38 | 444 | 46592 | 4.34 | 3.78 | 4.19 | 14.4 | 260 | 6304 | 4.78 | 4.62 | 22.4 | 79 | 185 | 378 |
| MambaExtend | 7.06 | 6.18 | 5.03 | 4.84 | 5.16 | 5.72 | 4.31 | 3.78 | 3.48 | 3.62 | 4.81 | 6.93 | 4.59 | 3.95 | 3.89 | 4.25 | 5.56 | 5.00 |

Table 2: Mamba vs MambaExtend performance on representative LongBench tasks.

| Model | Qasper | HotpotQA | 2WikiMultihopQA | TREC | TriviaQA | LCC | RepoBench-P | Average |
|---|---|---|---|---|---|---|---|---|
| Mamba-1.4B | 7.0 | 11.00 | 9.75 | 29.00 | 1.67 | 20.12 | 11.67 | 12.88 |
| MambaExtend-1.4B | **16.67** | **14.29** | **13.82** | **35.0** | **7.67** | **26.12** | **18.84** | **18.91** |
| Mamba2-780M | 7.50 | 6.06 | 9.48 | 17.0 | 0.1 | 22.1 | 14.01 | 10.89 |
| MambaExtend2-780M | **7.96** | **10.95** | **18.33** | **28.00** | **6.83** | **28.27** | **17.71** | **16.86** |

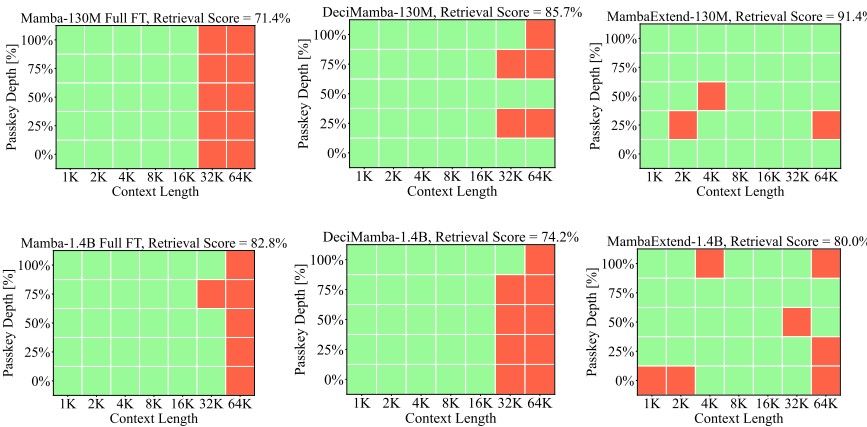

Figure 5: Passkey retrieval performance after fine-tuning (FT) (for Mamba and DeciMamba) or calibrating (for MambaExtend) on samples of 4k context length.

$\mathrm{CF}_{ZO}$ calibration function to learn the scaling factors. Similar to the calibration setup for perplexity evaluation, we calibrate one scaling factor per layer shared over the whole $\Delta_t$ tensor for that layer. As demonstrated in the Table 2, **MambaExtend can improve the average LongBench accuracy by up to** $6.03\%$.

**Passkey Retrieval**. Previous works have demonstrated that tasks requiring exact retrieval are more challenging than achieving low perplexity in longer context (Liu et al., 2024), so we use more fine-grained sharing of scaling factors to optimize. For $\Delta_t$ tensor of a layer $i$, we use one scaling factor per channel yielding total $D$ scaling factors per layer ($s_i \in \mathbb{R}^D_+$). Unless otherwise stated, we use $\mathrm{CF}_{BP}$ for *one* epoch to calibrate on a dataset with 4k context length. For the baseline, we performed standard fine-tuning with the same context length for one epoch as we get significant failure in the retrieval. For DeciMamba to have a fair comparison, we fine-tune for the same epochs as ours[4].

The evaluation is conducted across context lengths of 1K, 2K, 4K, 8K, 16K, 32K, and 64K, with the target digit hidden at depths of 0%, 25%, 50%, 75%, and 100% of each of these sequence. Assuming that each correct retrieval receives a score of 1 and each incorrect retrieval receives a score of 0, we compute the *retrieval score* in percentage (%) as $\frac{\text{Total correct retrievals}}{\text{Total (correct + incorrect) retrievals}} * 100$, across all the depths overall context lengths. The result is demonstrated in Fig. 5. Although MambaExtend calibrates approximately **3500× and 7100× fewer parameters for Mamba-130M and Mamba-1.4B, receptively, it performs better or very similarly to the other two alternatives**.

### 5.3 COMPUTE, TIME, AND MEMORY COST ANALYSIS

Fig. 6 demonstrates a comparison of full finetuning of baseline Mamba, DeciMamba, and calibration tuning with MambaExtend for the passkey retrieval task. Note here that to have a fair comparison and to demonstrate efficacy at extreme lost cost tuning, we set the epoch to one for all. For

---

[4]In the original paper (Ben-Kish et al., 2024) the model was fine-tuned for longer duration, however we focus on limited resource calibration and thus keep our experiments limited to fine-tuning for one epoch. Please see Appendix for fine-tuning results with longer epochs.

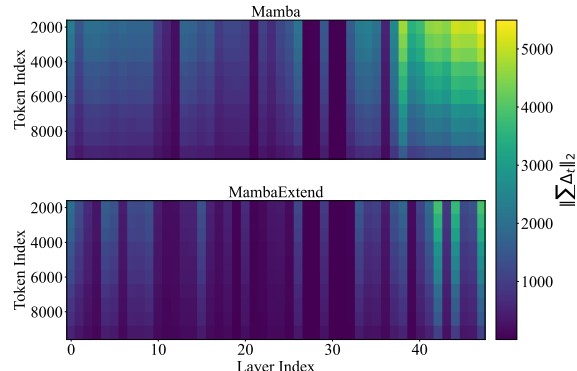

Figure 6: Comparison of normalized {*peak memory, calibration time, and number of parameter updates*} between Mamba, DeciMamba, and MambaExtend for passkey retrieval task. We use Mamba-130M model and for each method, we train for *one* epoch either with 4k or with 8k context length. For these three measurement types, we normalize each value by the corresponding value of MambaExtend-130M-4k.

MambaExtend, we show results for fine-tuning with both 4k and 8k contexts, while for others, we only perform experiments with tuning with 4k contexts. Notably, **MambaExtend requires up to $2.12\times$ fewer memory for tuning with similar context; in other words, it can support calibration with higher context of up to $2\times$.** Regarding per epoch calibration time, MambaExtend can be faster by up to $1.69\times$ while requiring up to $3532.6\times$ fewer parameters to update. To measure the retrieval success, we compute the Interestingly, despite having significant calibration efficiency, 4k tuned MambaExtend provides up to $20\%$ improved accuracy. We yield even better efficiency for $\text{CF}_{ZO}$ based calibration. In specific, compared to DeciMamba, MambaExtend requires up to $\sim \mathbf{5.42 * 10^6}\times$ fewer parameter updates and costs up to $\mathbf{3.87}\times$ lower peak-memory (details provided in Appendix A.3).

## 5.4 DISCUSSION AND ABLATION STUDY

**Understanding the impact of learned scaling on $\Delta_t$.** To understand the benefits of the learned scaling on the $\Delta_t$ discretization tensor, we compute the normalized sum of $\Delta_t$ $\|(\sum_{t=n}^{P'} \Delta_t)\|_2$. Here, $n$ refers to the token index whose impact we want to study on the output context length $P'$. $P'$ is set to 32k for this analysis. The Fig. 7 demonstrates the heatmap of the $\|(\sum_{t=n}^{P'} \Delta_t)\|_2$ for different token index ($n$) at different layers of the model. Notably, as discussed earlier, high $\|(\sum_{t=n}^{P'} \Delta_t)\|_2$ value may be associated with a stronger decaying effect on the output token $P'$. As we can see, the original Mamba, particularly for later layers, induces a significant decaying effect for the earlier tokens (see the value for token index 2000 for layer index $> 40$). This finding aligns with that of Ben-Kish et al. (2024). MambaExtend, on the contrary, reduces this effect significantly, both overall and for later layers.

Figure 7: Impact of the calibrated scaling factors on $\Delta_t$. (Top) layer-wise normalized sum of $\Delta_t$ for a pretrained Mamba. (Bottom) layer-wise Normalized sum of $\Delta_t$ layer-wise for a MambaExtend calibrated model. We used Mamba-1.4B on Pile with 32K context.

**Ablation on the granularity of scaling factor sharing.** Table 3 presents the results with various levels of sharing of the scaling factor a layer's $\Delta_t$. Specifically, we allow per-channel, per-token, and per-tensor sharing where a scaling factor is shared over a channel, a token, and the whole tensor for a layer's $\Delta_t$, respectively. We calibrate for one epoch for three scenarios. As we can see from the

Table 3: Passkey retrieval on Mamba-130M with a different granularity of the scaling factor sharing.

| Sharing granularity | # Params. ↓ | Retrieval Score (%) ↑ |
|---|---|---|
| Per-channel | 36.8K | **91.4** |
| Per-token | 98.3K | 62.8 |
| Per-tensor | **24** | 22.8 |

table, per-channel sharing can improve the retrieval score significantly. While per-tensor sharing requires considerably fewer calibration parameters, it fails to yield a good score, making per-channel sharing an optimal choice.

**Ablation on $CF_{BP}$ vs. $CF_{ZO}$.** For simpler long-context understanding tasks we demonstrated $CF_{ZO}$ to yield significantly improved PPL. In Table 4, we now demonstrate a direct comparison of the two calibration functions, namely, $CF_{ZO}$ and $CF_{BP}$ for Pile dataset with Mamba-130M. As we can see, the perplexities for the three evaluation context

Table 4: Perplexity result with $CF_{ZO}$ v.s. $CF_{BP}$ on Pile dataset.

| CF\Context Length | 4K | 8K | 16K |
|---|---|---|---|
| $CF_{BP}$ | **6.10** | 5.11 | **4.79** |
| $CF_{ZO}$ | 6.18 | **5.03** | 4.84 |

lengths are similar for both of these methods. This experiment demonstrates the efficacy of $CF_{ZO}$ along with its efficiency over $BP$-based alternative.

# 6 RELATED WORK

**Long-context understanding for LLMs**. (Chen et al., 2023a) introduced positional interpolation to mitigate the issue of OOD positions for contexts exceeding the pre-training length in RoPE-based transformers. In parallel, works such as (Han et al., 2024; Jin et al., 2024) proposed zero-shot techniques that constrain positional indices to discrete integer values when handling extended contexts in transformers. Additionally, (Chen et al., 2024) employs evolutionary search to design a non-uniform position interpolation and initialization strategy for fine-tuning on longer contexts. The YaRN method (Peng et al., 2024) further advances this line of work by combining positional interpolation with dynamic NTK-aware scaling, which dynamically adjusts the scaling of high- and low-frequency components of positional embeddings based on sequence length. Despite significant progress in transformer-based LLMs, long-context understanding for SSMs it yet to be fully unveiled. Only recently, inspired by the success of LongLoRA Chen et al. (2023c), DeciMamba (Ben-Kish et al., 2024) has proposed a fine-tuning-based context-extension for pre-trained models.

**Zeroth-order optimization**. Zeroth-order (ZO) optimization refers to a class of optimization algorithms that do not have backpropagation-based gradient computation. Instead, the ZO methods estimate gradients indirectly by querying function values through only forward passes. Over the past years, several techniques have been developed for ZO gradient estimation. Randomized Gradient Estimation (RGE) (Nesterov & Spokoiny, 2017) approximates gradient by randomly perturbing the input in multiple directions and examining the function value change. The perturbation is typically drawn from a random distribution, such as Gaussian or Rademacher. It potentially requires fewer function evaluations than alternative methods like finite differences (FD) (Shi et al., 2021). Simultaneous perturbation stochastic approximation (SPSA) (Spall, 1992) is a highly efficient ZO method for minimizing multivariate loss functions. Unlike the RGE and FD method, which requires multiple evaluations per iteration, SPSA perturbs all input dimensions simultaneously, requiring only two function evaluations per iteration, regardless of the problem's dimensionality. This makes SPSA especially attractive for large-scale optimization tasks. Recently, various algorithms, including MeZO (Malladi et al., 2023a), further improved the memory efficiency of SPSA. In the MambaExtend, we gain efficiency benefits by optimizing a small number of parameters.

# 7 CONCLUSIONS

We addressed the limitations of Mamba in handling long-context tasks by introducing MambaExtend, a framework to extend the context length of Mamba models without model training. Through non-uniform calibration of the discretization matrix ($\Delta_t$) scaling factors across different model layers, we enabled context extension by up to $32\times$ while maintaining similar perplexity levels. We believe this work opens up new possibilities for efficient, training-free adaptation of state-space models to long context applications, potentially allowing the true potential of sub-quadratic models to be unveiled. We hope our findings will inspire the community to delve further into the global-local ERF (Xiao et al., 2024) aware tuning, as well as the theoretical underpinning of the relation between discretization steps and OOD generalization.

# 8 ACKNOWLEDGMENTS

We would like to acknowledge the constructive discussions and feedback from the anonymous reviewers of the International Conference of Learning Representation 2025. This research was supported in part by a grant from the National Science Foundation.

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

# A  APPENDIX

## A.1  DETAILED HYPERPARAMETERS

**CF$_{ZO}$ hyperparameters.** For Pile, PG-19, and LongBench dataset calibration, we set the ZO optimization hyperparameters to $\eta = 0.001$, $c = 0.1$, and $K = 50$.

**CF$_{BP}$ hyperparameters.** For the passkey retrieval task, we train the models for one epoch using Adam optimizer with a learning rate of $0.1$ for MambaExtend. For DeciMamba, and full fine-tuning, we use the learning rate to be $1e - 4$, as suggested by the authors Ben-Kish et al. (2024). For all three cases, we use a batch size of 32, a gradient clipping of 1.0, a weight decay of 0.1, and train on a sequence length of 6144.

## A.2  PRE-TRAINED MODEL CHECKPOINTS USED

The pre-trained model checkpoints of Mamba are taken from the Hugging Face model Hub[5]:

- `state-spaces/mamba-130m`
- `state-spaces/mamba-1.4b`
- `state-spaces/mamba2-780m`

## A.3  MORE RESULTS

Fig. 8 shows the performance comparison of DeciMamba and MambaExtend in terms of compute, memory, and time. For DeciMamba, we use the total training time of 5 epochs, to evaluate the normalized FT time. For MambaExtend, as we use ZO for the calibration, we report the time associated with the 50 iterations of calibrations. Notably, for MambaExtend, we calibrate separately for each evaluation context length, while DeciMamba does one fine-tuning for 5 epochs with 2k

---

[5]https://github.com/state-spaces/mamba

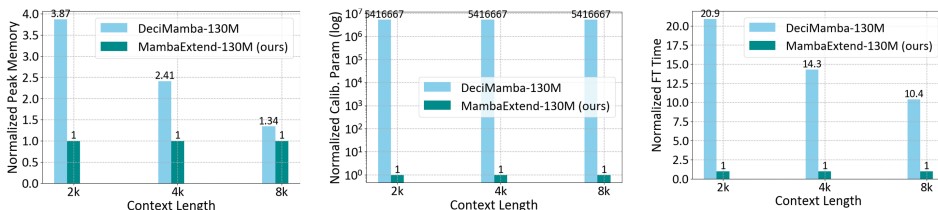

Figure 8: Comparison of normalized {*peak memory, number of parameter updates, and calibration/fine-tuning (FT) time (total)*} between DeciMamba, and MambaExtend for PG-19. We use Mamba-130M model for this evaluation.

Table 5: PPL comparison with transformer-based LLM for long-context understanding on Pile.

| Model | 2K | 4K | 8K | 16K | 32K | 64K |
|---|---|---|---|---|---|---|
| TinyLLaMA1.1B (2K) | **4.6** | 62.6 | 426.6 | 1243.7 | 2684.6 | 3372.04 |
| TinyLLaMA1.1B-PI | **4.6** | 9.56 | 50.34 | 116.47 | 168.84 | 229.46 |
| MambaExtend-130M | 7.06 | **6.18** | **5.03** | **4.84** | **5.16** | **5.72** |

context length. This causes the peak memory and fine-tuning time to increase for MambaExtend while keeping them constant for DeciMamba. For each evaluation metric, we normalize the corresponding value for MambaExtend at the context length under consideration. As Fig. 8 shows, MambaExtend requires $\sim 5.42 * 10^6 \times$ fewer parameter updates and costs up to $3.87 \times$ lower peak memory. Additionally, MambaExtend provides up to $20.9 \times$ faster calibration as opposed to the fine-tuning duration of DeciMamba.

## A.4 COMPARISON WITH TRANSFORMER-BASED LLMS

Supporting longer contexts during inference is a critical challenge for both transformer-based and Mamba-based LLMs. In this section, we compare our performance with transformer-based models to provide a broader perspective on the long-context extension results achieved with Mamba models. Specifically, we use the TinyLLaMA-1.1B model, which is trained on a 2K context length, as our baseline transformer

Table 6: PPL comparison with transformer-based LLM for long-context understanding on PG19.

| Model | 16K | 32K | 64K |
|---|---|---|---|
| TinyLLaMA1.1B (2K) | 2236.98 | 4205.64 | 8664.11 |
| TinyLLaMA1.1B-PI | 226.69 | 300.46 | 375.49 |
| MambaExtend-130M | 19.25 | 20.3 | 25 |
| MambaExtend-1.4B | **14** | **14.34** | **16.12** |

model. We also consider a positionally interpolated version of the same model. It is important to note that positional interpolation (PI) is a widely used training-free method for extending the context of transformer models. As shown in Table 5, the MambaExtend model, despite being smaller, consistently outperforms TinyLLaMA-1.1B at longer context lengths, both with and without PI. Furthermore, we compare the performance of TinyLLaMA-1.1B (with and without PI) and MambaExtend on PG19, another popular benchmark for perplexity (PPL) evaluation on long contexts. The results, presented in Table 6, clearly demonstrate the significant performance advantages of MambaExtend compared to transformer-based alternatives. It is noteworthy that MambaExtend significantly surpasses the TinyLLaMA model variants, whether they are smaller or of similar size, highlighting its efficacy.

## A.5 MORE COMPARISON WITH DECIMAMBA

While in the main manuscript, we demonstrate the benefits of MambaExtend over the baseline Mamba on Pile dataset, we now show a comparison with DeciMamba (Ben-Kish et al., 2024) on the same. In specific, Table 7 demonstrates the efficacy of MambaExtend in maintaining the PPL better than DeciMamba, particularly at longer contexts with context length $\geq$ 8K. Additionally, we show

Table 7: PPL comparison between DeciMamba and MambaExtend on Pile.

| Model | 2K | 4K | 8K | 16K | 32K | 64K |
|---|---|---|---|---|---|---|
| DeciMamba-130M | **4.93** | **5.36** | 5.21 | 6.99 | 8.19 | 10.62 |
| MambaExtend-130M | 7.06 | 6.18 | **5.03** | **4.84** | **5.16** | **5.72** |

results on LongBench to compare with that generated by DeciMamba in a zero-shot fashion. In

Table 8: F1 scores on HotpotQA and Qasper from LongBench on DeciMamba and MambaExtend, respectively. *Italicized* numbers identify the results taken from (Authors LongMamba, 2024) paper.

| Model | HotpotQA | Qasper |
|---|---|---|
| DeciMamba-1.4B | *13.88* | *14.24* |
| MambaExtend-1.4B | **14.29** | **16.67** |

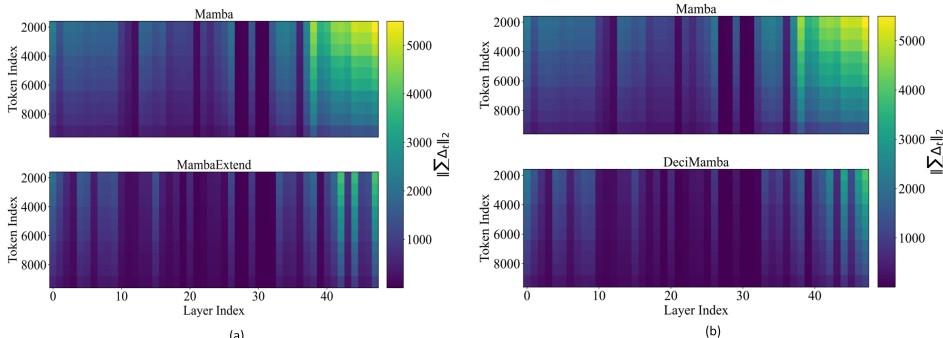

Figure 9: Impact of the calibrated scaling factors on $\Delta_t$ (a) Mamba vs. MambaExtend, and (b) Mamba vs. DeciMamba as evaluated on Pile 32K context length. (Top) of both (a) and (b) shows layer-wise Normalized sum of $\Delta_t$ for a pre-trained Mamba-1.4B. (Bottom) layer-wise Normalized sum of $\Delta_t$ for (a) MambaExtend-1.4B calibrated model, and (b) DeciMamba-1.4B fine-tuned model. To fine-tune DeciMamba 1.4B model we adhered to the setup described in (Ben-Kish et al., 2024).

specific, Table 8 shows that MambaExtend can yield reasonably improved performance as evaluated on HotpotQA and Qasper, respectively.

**Comparing the impact of learned scaling and full fine-tuning on $\Delta_t$.** MambaExtend employs a learned scaling policy to adjust the discretization steps, $\Delta_t$. In contrast, DeciMamba (Ben-Kish et al., 2024) performs full model fine-tuning to enhance performance on longer contexts. We now visualize the impact of these two approaches on the normalized sum of $\Delta_t$ per layer. Specifically, in Fig. 9, we present a direct comparison of MambaExtend (9(a)) and DeciMamba (9(b)). Interestingly, both approaches exhibit a similar influence on the normalized sum of $\Delta_t$'s, significantly reducing its values in the later layers. This experiment demonstrates that both methods effectively recalibrate the $\Delta_t$'s. However, our approach achieves similar benefits with greater computational and memory efficiency and reduced latency.

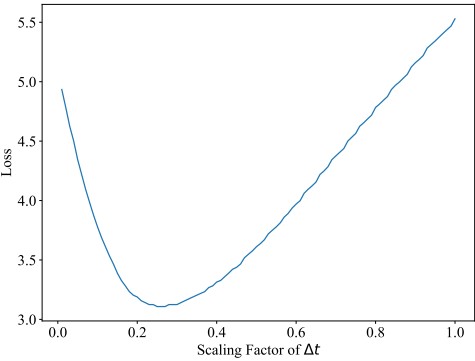

Figure 10: : Impact of different values of uniform $\Delta_t$ scaling on the loss landscape of the model.

## A.6 THE LOSS LANDSCAPE FOR GRID-SEARCHED SCALING FACTORS

Fig. 3 in the main manuscript demonstrates the impact of uniform $\Delta_t$ scaling per layer in terms of PPL value. We now plot the loss landscape of the model with uniform scaling factor values in the same range as that of Fig. 3. In specific, 10 shows the loss landscape to have a convex nature as we sweep over the scale factors ($s$) in $0 < s \leq 1$.

Table 9: PPL comparison with TBTT (Wang, 2024) fine-tuned model on Pile.

| Model | TBTT fine-tuned | 4K | 8K | 16K | 32K | 64K |
|---|---|---|---|---|---|---|
| Mamba2-780M (baseline) | No | **4.62** | 22.4 | 79 | 185 | 378 |
| Mamba2-780M | Yes | 4.62 | 4.34 | **3.89** | **4.92** | 5.16 |
| Mamba2Extend-780M | No | **3.95** | **3.89** | 4.25 | 5.56 | **5** |

Table 10: Comparison with TBTT (Wang, 2024) fine-tuned model on Passkey retrieval task.

| Model | TBTT fine-tuned | Avg. Accuracy (%) |
|---|---|---|
| Mamba2-780M (baseline) | No | 0 |
| Mamba2-780M | Yes | 5.7 |
| Mamba2Extend-780M | No | **91.34** |

Table 11: Comparison between fine-tuning and calibration for longer epochs on Passkey retrieval.

| Model | Passkey retrieval acc. (%) |
|---|---|
| DeciMamba-130M | 93.1 |
| MambaExtend-130M | **94.3** |

## A.7   COMPARISON WITH MODELS FINE-TUNED VIA TRUNCATED BACKPROPAGATION THROUGH TIME

Contemporary research on Mamba2 models trained using truncated backpropagation through time (TBTT) has demonstrated promising generalization capabilities on longer contexts (Wang, 2024). To compare MambaExtend with the TBTT fine-tuned model, we apply fine-tuning for three epochs based on the TBTT approach on a pre-trained Mamba2-780M using 0.8B tokens from the PG19 train split. We then evaluate performance on the Pile and Passkey retrieval tasks and present our comparisons with MambaExtend in Tables 9 and 10, respectively. Interestingly, we observe a notable performance improvement for the Pile dataset for longer contexts, nearing the performance of MambaExtend. However, in the critical benchmark of long-context retrieval (Table 10), the TBTT fine-tuned model exhibits negligible retrieval accuracy. In contrast, MambaExtend achieves a significant accuracy boost solely by calibrating the scaling factors.

**Important Notes on TBTT Training.** The TBTT-based fine-tuning method shares similarities with the DeciMamba approach (Ben-Kish et al., 2024), which also proposes full fine-tuning to enhance long-context understanding (albeit without TBTT). However, we emphasize that the key advantage of scaling-based calibration in MambaExtend is that it is an orthogonal method compared to such full fine-tuning approaches. This not only leads to improved accuracy but also offers significant computational and memory benefits, potentially enabling calibration in resource-constrained environments. Furthermore, as illustrated in Figure 7 of the original LongSSM paper (Wang, 2024), training relatively large models, such as the 140M S5, with a previously initialized state (under the TBTT policy) can lead to severe stability issues. This raises concerns about the scalability of such approaches, as identified by the authors.

## A.8   FINE-TUNING VS. CALIBRATION FOR LONGER EPOCHS

Table 11 shows results of fine-tuning with DeciMamba for five epochs on passkey retrieval. For a fair comparison, we show the results of MambaExtend with scaling factors calibrated for the same epochs. As we can see, MambaExtend can still retain improved performance over the other. However, please note, in this work we aim to achieve long context generalization with minimal compute and calibration overhead, thus we aim to focus on fine-tuning for only one epoch.

## A.9   HARDWARE AND API RESOURCES USED

We used an Nvidia A6000 GPU with 48 GB memory for all the experiments. To perform calibration and fine-tuning, we used Pytorch API to write the corresponding code.

