# OpenReview forum: "MambaExtend: A Training-Free Approach to Improve Long Context Extension of Mamba"
_ICLR.cc/2025/Conference — ICLR 2025 Poster_

### Official Review · Reviewer_UZhN · 2024-10-29

**Soundness:** 3
**Presentation:** 2
**Contribution:** 3
**Rating:** 6
**Confidence:** 4

**Summary:**

The paper proposes MambaExtend a computationally efficient way to improve Mamba's ability handle longer contexts.
The method is based on the observation that when presented with longer sequences than during training, the time discretization parameter of Mamba assumes values which the model cannot handle.
As a result, the authors propose to use a similar approach to position interpolation in transformers and scale the discretization parameter per layer independently to a range which Mamba has seen during training.
The authors propose two methods to estimate the scaling factor for the discretization parameter of each layer a gradient-based and a zero-order optimization method using finite-differences to estimate the scaling factor and ensure that the scaling factor can only be positive.
The authors demonstrate improved performance on pass-key retrieval tasks and show significantly lower perplexity at high context lengths.

**Strengths:**

- The proposed method allows Mamba to be used with much higher context lengths than seen during training. Given that long-context handling is the computational advantage of Mamba models, it's great that the proposed method allows to alleviate performance degradation.
- The results on pass-key retrieval are convincing.
- The method is well-motivated and easy to understand.
- The authors propose two methods: a more accurate but also more memory heavy gradient based method and a less accurate but less memory intensive zero-order method which makes sense to me as they are applicable depending on the model size and hardware constraints.

**Weaknesses:**

- While the proposed method is effective at making an existing checkpoint of Mamba more capable of handling long contexts, I feel like it treats more a symptom rather than a cause. As shown in Wang et al. (2024) (Figure 4) Mamba can handle long context effectively if trained using truncated backpropagation through time on consecutive chunks of the data (rather than shuffling). Crucially, the initial state of Mamba is changed from zero to the last hidden state in the sequence. As a result Mamba doesn't need to start from scratch at token position i every time.
This is challenging to implement in Mamba 1 as the initial state can't be set but is an exposed option for Mamba 2. I think it would be interesting to understand whether the observed differences in the discretization parameter for longer context still occur in models trained in this way or whether it makes the proposed method less effective. This could be checked experimentally using an existing small checkpoint (340m) and then fine-tuning on a small dataset of around 1B token using TBTT.
- The code for reproducibility was not provided.




References:
- Wang, S. (2024). LongSSM: On the Length Extension of State-space Models in Language Modelling. arXiv preprint arXiv:2406.02080.

**Questions:**

- Please see the main weakness. In a sense, I think it would be important to understand whether the performance degradation you observe for Mamba is an artifact of the training using a transformer pipeline where the data is shuffled into non-consecutive chunks compared to an RNN training pipeline where the data isn't chunked and the last hidden state of the previous batch is the first hidden state of the current batch.

---

### Official Review · Reviewer_rYkc · 2024-11-04

**Soundness:** 3
**Presentation:** 3
**Contribution:** 3
**Rating:** 8
**Confidence:** 3

**Summary:**

This paper aims to address the length extrapolation problem in Mamba. The authors are inspired by the $\Delta_t$ discussion in DeciMamba and further investigate the role of $\Delta_t$, finding that the longer sequence will have stronger forgetting for the previous sequence. Thus, they propose an approach to adjust the $\Delta_t$ for length extrapolation. The proposed method is smooth to follow and the 2 different optimization methods provide more computationally efficient solutions.

**Strengths:**

1. Originality: this paper further investigates the role of $\Delta_t$ in Mamba length extrapolation, and proposes an easy but novel algorithm to learn the scaling of $\Delta_t$ using both first and second-order methods. Provided a computationally efficient and general method for the targeted problem.
2. Quality and Clarity: writing is easy to follow, and experiments are comprehensive and clear.
3. Significance: this work is like PI in a normal transformer and should be a significant contribution.

**Weaknesses:**

1. Experiments are not fully compared with existing work (e.g. DeciMamba). In Tables 1 and 2, the authors only compare with the vanilla Mamba.
2. Figure 5: the passkey experiment, is different from the DeciMamba[1] Figure 8 and LongMamba[2] Figure 3, where DeciMamba performs better compared to Fig 5's result. Could authors explain the possible reason for the difference?

[1] DeciMamba: Exploring the Length Extrapolation Potential of Mamba

[2] LONGMAMBA: ENHANCING MAMBA’S LONGCONTEXT CAPABILITIES VIA TRAINING-FREE RECEPTIVE FIELD ENLARGEMENT

**Questions:**

See Weakness

---

### Official Review · Reviewer_SxC7 · 2024-11-06

**Soundness:** 2
**Presentation:** 2
**Contribution:** 2
**Rating:** 6
**Confidence:** 2

**Summary:**

This paper introduces MambaExtend, an approach to improve the long-context capabilities of Mamba models.
Key contributions are:

The authors leverage the out-of-distribution discretization steps to improve Mamba's poor generalization to longer contexts. They show that scaled-down discretization steps can improve performance on longer sequences.
They propose MambaExtend, which introduces learnable scaling factors for the discretization steps (Δt) in each layer while keeping the pre-trained model weights frozen. The framework includes both gradient-based and gradient-free (zeroth-order) optimization approaches to learn these scaling factors.
With their method, they enable context extension up to 32× (from 2k to 64k tokens) without significant perplexity degradation
and achieve up to 40.6% reduced perplexity compared to alternative approaches.

**Strengths:**

The paper is quite well writtin and easy to follow. The authors propose a simple approach to scale the out-of-context generalization of Mamba by tuning the discretization steps of the architecture. The method is intutitve and quite well analysed.

**Weaknesses:**

I find the term "training-free" quite misleading. Afaiu the method does train the discretization steps, with backprop or approximation. Please clarify what you mean with this.

I find it also hard to judge what is novel in this work compared to the (recent) DeciMamba, in terms of insights, as they already, to the best of my knowledge, identify the discretization steps to be crucial for solving longer context generalization. Generally, I do not find these insights surprising.

**Questions:**

Figure 3 indicites, as its also quite intutitive, that line searching over the different scaling factors would be sufficient - althought potentially costly. Generally, it would be interesting to analyse the loss landscape wrt to the discretization steps in general and confirm its convex nature.

Can you highlight more the differences to DeciMamba, and work on showing qualitative / quantiative results. Afaiu DeciMamba tunes all weights, and therefore induces more memoery and compute costs? Does GD on all weights nevertheless tunes (primarily) the discretization steps, nevertheless? This could be analysed quite easily imo and would highlight your contribution, showing that GD on all parameters anyway tunes what you propose to tune.

---

### Official Review · Reviewer_jd9z · 2024-11-09

**Soundness:** 2
**Presentation:** 3
**Contribution:** 2
**Rating:** 5
**Confidence:** 3

**Summary:**

This paper introduces MambaExtend, a novel framework designed to enhance the long-context capabilities of Mamba models, a leading type of state-space model (SSM) known for achieving impressive speed in language mode traininig/inference. this paper identifies the limitations of Mamba in generalizing to longer contexts due to out-of-distribution discretization steps and proposes a training-free approach to address this challenge. The authors have shown that their training-free method successfully increases the context length of Mamba models while preserving computing efficiency. This paper's primary finding indicates significant enhancements in perplexity scores on the Pile and PG-19 datasets, achieving increases of up to 8145x and 32506x, respectively.

**Strengths:**

* The authors empirically found that scaling down the discretization step size (Δt) can improve generalization on increased context lengths at inference time.

* This paper enhances Mamba's context length without retraining the model weights by leveraging a calibration function (CF) to optimize the discretization step sizes (Δt) across various Mamba layers, which significantly reduces the number of parameter updates and peak memory demand compared to traditional fine-tuning methods

**Weaknesses:**

* Limited Baseline Comparisons The study lacks comparisons with non-Mamba architectures like transformers on Pile and LongBench datasets; Including established techniques like Positional Interpolation would provide valuable context about relative performance improvements; A broader comparison framework would better highlight MambaExtend's contributions within the larger landscape of long-context models

* Insufficient Coverage of Related Work: Lack of discussion of non-mamba techniques that addressing the long context LLM techniques, such as  NTK-Aware scaling, LongLora. A discussion of how these existing approaches compare or could complement MambaExtend would strengthen the paper's positioning

* Deeper Explanation of Discretization and Its Role in OOD: While the paper states that OOD discretization steps limit Mamba's long-context performance, it could benefit from a more thorough explanation of what discretization is, how it specifically relates to Mamba's S6 layers, and why it becomes problematic for longer sequences.  the paper mentions that scaling down Δt improves long-context generalization but doesn't fully explain the underlying rationale. While the paper demonstrates that reducing Δt improves performance, the theoretical underpinning and connection to effective receptive field (ERF) warrant deeper exploration.

* How to choice of CF is non clear: The rationale for choosing between CFZO and CFBP calibration functions lacks clarity, The paper mentions using CFZO for simpler tasks and CFBP for more complex ones due to the number of scaling factors involved. This could be confused, would you elaborating more detail, guideline of criteria of selecting the calibration function?

**Questions:**

see weaknesses above

---

### Meta-Review · Area_Chair_bTNe · 2024-12-22

**Metareview:**

Summary: MambaExtend is a novel framework that enhances the long-context capabilities of Mamba models, a type of state-space model known for its speed in language model training and inference. It addresses Mamba's limitations in generalizing to longer contexts by proposing a training-free approach that adjusts the discretization step sizes, leading to significant improvements in perplexity scores on the Pile and PG-19 datasets.

Strengths:

MambaExtend effectively increases the context length of Mamba models without retraining, leveraging a calibration function to optimize discretization steps and reduce computational resource demands.

The framework introduces an intuitive and well-analyzed method that achieves up to 40.6% reduced perplexity compared to alternative approaches, demonstrating its potential for energy-efficient and accurate graph-based systems.

Drawback:

The paper could benefit from a deeper theoretical explanation of the discretization process and its impact on out-of-distribution generalization, as well as a clearer rationale for choosing between different calibration functions.

Given the strengths and the potential for MambaExtend to significantly improve the performance of Mamba models in handling longer contexts, I accept this work as a borderline paper that contributes valuable insights and solutions to the field.

**Additional Comments On Reviewer Discussion:**

Most of the concerns have been well-addressed.

---

### Decision · Program_Chairs · 2025-01-22

Accept (Poster)